# Compact Wideband Groove Gap Waveguide Bandpass Filters Manufactured with 3D Printing and CNC Milling Techniques

**DOI:** 10.3390/s23136234

**Published:** 2023-07-07

**Authors:** Clara Máximo-Gutierrez, Juan Hinojosa, José Abad-López, Antonio Urbina-Yeregui, Alejandro Alvarez-Melcon

**Affiliations:** 1Department of Information and Communications Technology, Universidad Politécnica de Cartagena, Plaza del Hospital no. 1, 30202 Cartagena, Spain; clara.maximo@upct.es; 2Department of Electronics and Computer Engineering, Universidad Politécnica de Cartagena, Plaza del Hospital no. 1, 30202 Cartagena, Spain; antonio.urbina@upct.es (A.U.-Y.); alejandro.alvarez@upct.es (A.A.-M.); 3Department of Applied Physics, Universidad Politécnica de Cartagena, Calle Doctor Fleming s/n, 30202 Cartagena, Spain; jose.abad@upct.es

**Keywords:** 3D printing, bandpass filter, CNC machining, groove gap waveguide technology, lowpass filter, stepped impedance synthesis

## Abstract

This paper presents for the first time a compact wideband bandpass filter in groove gap waveguide (GGW) technology. The structure is obtained by including metallic pins along the central part of the GGW bottom plate according to an *n*-order Chebyshev stepped impedance synthesis method. The bandpass response is achieved by combining the high-pass characteristic of the GGW and the low-pass behavior of the metallic pins, which act as impedance inverters. This simple structure together with the rigorous design technique allows for a reduction in the manufacturing complexity for the realization of high-performance filters. These capabilities are verified by designing a fifth-order GGW Chebyshev bandpass filter with a bandwidth *BW* = 3.7 GHz and return loss *RL* = 20 dB in the frequency range of the WR-75 standard, and by implementing it using computer numerical control (CNC) machining and three-dimensional (3D) printing techniques. Three prototypes have been manufactured: one using a computer numerical control (CNC) milling machine and two others by means of a stereolithography-based 3D printer and a photopolymer resin. One of the two resin-based prototypes has been metallized from a silver vacuum thermal evaporation deposition technique, while for the other a spray coating system has been used. The three prototypes have shown a good agreement between the measured and simulated *S*-parameters, with insertion losses better than *IL* = 1.2 dB. Reduced size and high-performance frequency responses with respect to other GGW bandpass filters were obtained. These wideband GGW filter prototypes could have a great potential for future emerging satellite communications systems.

## 1. Introduction

Microwave filters are key devices in the communications payload of satellites [1]. Among different technologies, a rectangular waveguide (RW) is often employed to implement them at microwave/mm-wave frequencies, due to its low loss, high quality factor, and high-power handling capability with regards to planar and substrate integrated waveguide (SIW) solutions [2,3,4,5,6,7]. However, unlike RW technology, planar and SIW filters have a low profile and weight, small sizes, and allow the easy integration of active components. Moreover, low-cost planar and SIW filter prototypes can be manufactured using printed circuit technology. Their main disadvantages compared to RW filters are higher insertion losses (*ILs*), due to the dielectric substrate inserted in the structure; the small thickness of the substrate; and the dimensions and lower conductivity of the conductor type employed.

Recently, many modified SIW technologies have been proposed to improve insertion losses by partially or fully removing the dielectric substrate. The structures with the fully removed substrate are named empty and dielectricless SIWs (ESIW and DSIW) [8,9], while those with the substrate partially eliminated are called air-filled, hollow, and modified SIWs (AFSIW, HSIW, and MSIW) [10,11,12]. Among these different structures, the empty substrate integrated waveguide (ESIW) has the lowest insertion loss and exhibits propagation characteristics analogous to the RW [13,14]. Thus, several filters with small and large fractional bandwidths (*FBWs*) have been successfully implemented in ESIW technology in recent years to benefit from RW-like properties. The designs with *FBW* lower than 4% are based on coupled cavities, resonators, and inductive posts [8,15,16,17,18,19], while *FBWs* higher than 10% are obtained by means of stepped impedances [19]. These ESIW filters have the advantages of planar and SIW technologies. However, they still present smaller quality factors than RW technology. In general, these filters are manufactured in two pieces with top and bottom parts. The ESIW and RW technologies have the same manufacturing drawback, since they require a good electrical contact between the top and bottom parts to avoid leaky waves. To overcome this problem in ESIW devices, an improved fabrication process was proposed in [20]. Nevertheless, this technique requires many plated vias to be drilled around the whole structure and this can be complex and expensive for the development of compact ESIW filters.

Gap waveguide (GW) technology was proposed in [21] to enhance the wave propagation in the air gap along the longitudinal direction of two parallel metallic surfaces. GW is based on two parallel plates and open sidewalls. One of the two parallel plates is composed of a conductive line centered between two periodic structures of metallic pins, which work as two artificial perfect magnetic conductor (PMC) sidewalls. The other one is a metallic plane, which behaves as a perfect electric conductor (PEC) surface. In this way, this structure eliminates the need for metallic contacts between the top and bottom parts, and the wave propagation takes place in the air groove area included between the two PMC sidewalls. These characteristics make this technology suitable for implementing millimeter and submillimeter band guided structures with low insertion loss and for manufacturing low-cost components. Based on the GW concept, a groove gap waveguide (GGW) was proposed in [22]. Recently, inverted microstrip and coplanar gap waveguides based on this idea were also presented [23,24,25]. However, the dielectric thickness in these new transmission lines negatively affects the losses with respect to the GGW. The GGW was designed as a potential alternative to the conventional RW at microwave/mm-wave frequency bands, but also it has a great potential in low-frequency applications (e.g., filters). GGW and RW have similar quality factor (*Q*) values and dispersive behaviors [26,27]. GGW requires the fabrication of two periodic structures of metallic pins acting as PMC sidewalls. These periodic structures slightly increase the complexity and fabrication cost, and the width of the waveguide with respect to classic waveguide technologies. However, these drawbacks are not enough compared to its benefits, since unlike classic waveguide technologies it does not need an electrical contact between the bottom and top plates, thus reducing the alignment problems for low- and high-frequency bands. Poor electrical contact between metallic blocks in classic waveguide technologies, with a cut in the E-plane, can produce high losses, thus deteriorating the *Q*-factor in low- and high-frequency applications (e.g., filters). Recently, glide-symmetric holey electromagnetic bandgap (EBG) structures have also been proposed instead of periodic structures of metallic pins to perform artificial PMC sidewalls for cost-effective gap waveguide technology [28,29].

Several bandpass filters have been successfully implemented in GGW technology [30,31,32,33,34,35,36,37,38,39,40,41,42,43,44,45]. Their designs are based on resonant cavities [30,31,32,33,34,35,36], coupled resonators [37,38,39,40,41,42], evanescent wave coupled resonators [43] and periodic corrugations acting as resonators [44]. Due to their configurations, these bandpass filters are narrowband type. Their frequency responses have an *FBW* lower than 4%. A wideband bandpass filter with *FBW* = 22.2% has also been proposed, introducing quasi-periodic transverse corrugations inside the GGW [45]. The resulting structures have been manufactured using computer numerical control (CNC) milling or 3D printing techniques and are generally small.

The aim of this paper is to design compact bandpass filters in GGW technology with large fractional bandwidths (*FBW* > 10%) using a robust technique against manufacturing tolerance, thus allowing their low-cost implementation with an emerging three-dimensional (3D) printing technique. To achieve compactness and large *FBWs*, simple metallic pins have been employed, and a rigorous synthesis method based on stepped impedances has been applied to obtain the desired frequency response. The synthesis technique uses Chebyshev transfer functions and impedance inverters [46,47], which are realized by means of suitably centered, distributed, and sized metallic pins along the groove area of the GGW. Bandpass response is realized by combining this rigorous low-pass synthesis technique with the high-pass characteristics (quasi-TE_10_ mode) of the GGW. The proposed design technique and the behavior of the pins inserted in this wideband GGW filter differ from the pin-based bandpass filter solutions proposed in [33,34,42,43]. The metallic pins employed in [33,34] act as couplings between the resonant cavities realized in the periodic structures (PMC). A modal analysis [33] and then a technique using space mapping and coupling matrix [34] were applied to design these bandpass filters. In [42], the metallic pins are periodically inserted along the groove area of the GGW filter and work as resonators or couplings. With this last structure [42], it is possible to achieve bandpass filters of different narrow fractional bandwidths (*FBWs*). However, the technique proposed in this work [42] does not allow us to accurately define the return loss (*RL*) and the bandwidth (*BW*) of the GGW bandpass filter. In [43], the pins act as resonators in an evanescent GGW section. With this structure, it is difficult to achieve large input couplings and, therefore, it is impossible to obtain a wide bandwidth (*FBW* > 10%). The slow wave solution presented in [45] allows the design of GGW bandpass filters with different bandwidths, using the cut-off frequency of the GGW and band-stop effect of the quasi-periodic transverse corrugations inside the GGW. However, this method requires a large number of corrugations to achieve a high level of rejection in the out-of-band region, which makes it difficult to determine the upper edge of the bandwidth and increases the size of the filter. Regarding manufacturing, this slow wave solution and the stepped impedance structure proposed in this work are more robust against tolerance than the bandpass filters using high *Q*-factors resonant elements. The structures that result from this design method are suitable not only for prototype manufacturing using a computer numerical control (CNC) machining technique, but they can also be fabricated with an emerging 3D printing technique [36,41,48], thus providing a new class of low-cost and low-weight components for communications systems.

In this work, a compact fifth-order wideband Chebyshev bandpass filter in GGW technology is proposed for the first time. The design of this filter with a bandwidth *BW* = 3.7 GHz (*FBW* = 36.4%) and return loss *RL* = 20 dB is described in Section 2. Section 3 is focused on the fabrication of three prototypes of this filter by means of a CNC milling machine and a stereolithography-based 3D printer to make a comparison between both fabrication techniques. The surfaces of two resin-based 3D prototypes were metallized using a silver vacuum thermal evaporation technique and a conductive paint spray coating system. The electromagnetic (EM) simulations and measurements of the three manufactured filter prototypes are presented and discussed in Section 4. In addition, the frequency response performance and size of these filters are compared with other bandpass filters in GGW technology. Finally, conclusions are presented in Section 5.

## 2. Design of the Wideband GGW Bandpass Filter

The structure of the proposed wideband Chebyshev bandpass filter based on GGW technology is depicted in Figure 1. It is a fifth-order stepped impedance filter. The bandpass response is obtained by combining the high-pass behavior of the GGW with the low-pass characteristics of the properly centered, distributed, and sized metallic pins along the groove area (*W*) of the GGW, which act as impedance inverters. The design steps of this bandpass filter according to the design specifications given in Table 1 are described below. According to the expression of the insertion loss defined in [49], high-order or very narrowband bandpass filters lead to higher lossy structures. Therefore, a fifth-order Chebyshev bandpass filter with *FBW* = 36.4% and *RL* = 20 dB was selected.

### 2.1. GGW Design

As shown in Figure 1, a vertically polarized GGW is used [22]. In this subsection, we consider the structure without the central metallic pins ($t_{xi}{\;\times \;t}_{yi}\;\times \;t_{zi}$, i=1,…,6), to emulate a standard empty rectangular waveguide. The bottom plate includes two periodic structures of three rows of square metallic pins (d × a) with pitch *p*, which are separated from each other by a groove width *W*. Both periodic structures act as artificial perfect magnetic conductor (PMC) sidewalls in a certain frequency band and do not allow wave propagation along the *x* and *y* transverse directions between the gap *h*. The upper plate is situated at a height b=d+h from the bottom part and realizes a perfect electric conductor (PEC) surface. In this way, by adjusting the dimensions of the periodic structure and the width *W* of the groove area, it is possible to propagate a quasi-$TE_{10}$ mode. Table 2 includes the optimized GGW dimensions that lead to the dispersion diagram of Figure 2, which was obtained with an electromagnetic (EM) simulator (HFSS) to achieve the high-pass characteristic of the GGW with a cut-off frequency $f_{C,L}=8.3$ GHz at $\left|S_{11}\right|=-3$ dB. As can be seen in Figure 2, the desired $TE_{10}$ mode in the GGW propagates along the *z*-axis, inside the stopband of the periodic structure, thus avoiding leakage in the transverse plane between 8.3 GHz and 15.5 GHz. The dimensions *W* and *b* of the GGW given in Table 2 match the WR-75 standard rectangular waveguide size.

### 2.2. Stepped Impedance Analysis

Figure 3 represents the structure and the equivalent circuit of the *i*th-section (i=1,…,6) of the fifth-order stepped impedance filter shown in Figure 1. Each section of the filter has a length $L_{i}$, which includes a metallic pin $t_{xi}{\;\times \;t}_{yi}\;\times \;t_{zi}$ centered on the width *W* of the GGW. The metallic pin works as an impedance inverter $K_{i}$, while the length $L_{i}$ realizes a phase shift of ${-\theta }_{e}-90{}^{\circ}$. Figure 4 depicts the simulated $\left|S_{21}\right|$-parameter of the *i*th-section of the filter structure as a function of the three dimensions ($t_{xi}$, $t_{yi}$, $t_{zi}$) of the metallic pin. The dimensions of the base GGW were the same as in Table 2. The centered metallic pins were considered square like those of the PMC sidewalls. The initial size of the centered metallic pin was fixed to that of the square metallic pin of the PMC sidewalls, except the height, which was selected at half: $t_{xi}=t_{zi}=a=2$ mm and $t_{yi}=d/2=4$ mm. As can be seen in Figure 4, the variation between 0.5 mm and 8 mm of the pin width ($t_{xi}$, $t_{zi}$) according to the *x* or *z* axis has little effect on the $\left|S_{21}\right|$-parameter, since its magnitude can only vary between 0.88 and 0.7. However, the behavior is different when the pin height $t_{yi}$ increases from 0.5 mm to 8 mm since the magnitude of $\left|S_{21}\right|$ decreases from 0.99 to 0.01. Therefore, to achieve the design specifications of the filter (Table 1), the height $t_{yi}$ of the metallic pin and the length $L_{i}$ of the six sections will be adjusted, while the widths $t_{xi}$ and $t_{zi}$ of all inverters will be fixed to 2 mm for practical considerations.

### 2.3. Filter Design

The design of the proposed fifth-order GGW bandpass filter (Figure 1) is performed by combining the high-pass behavior of the GGW with a low-pass Chebyshev filter synthesis method based on stepped impedances [46,47], thus allowing us to control its bandwidth. The lower cut-off frequency ($f_{c,L}$) of this filter was previously adjusted to the desired value (Table 1), optimizing the GGW dimensions as in Section 2.1. Following this, the synthesis method based on stepped impedances was applied to obtain the transfer function of the fifth-order Chebyshev filter according to the design specifications ($f_{c,H},$
*RL*, $\theta_{e}$) required by Table 1. The five characteristic impedances (Figure 5a) of the filter are extracted by applying an iterative algorithm to the [ABCD] matrix and the previously determined transfer function [47]. To achieve a practical implementation of the filter as in Figure 1, the five stepped impedance transmission lines ($Z_{i},i=1,\ldots ,5$) with electrical length ($\theta_{e}$) in Figure 5a are transformed into six sections (Figure 5b) of same impedance ($Z_{0}$) and electrical length ($\theta_{e}$) by means of impedance inverters ($K_{i},i=1,\ldots ,6$). In Figure 5b, the impedance inverter $K_{i}$ corresponds to the metallic pin $t_{xi}{\;\times \;t}_{yi}\;\times \;t_{zi}$ of the *i*th-section of the filter (Figure 3), while the electrical length $\theta_{e}$ is related with the physical section length $L_{i}$. The normalized impedance inverter $K_{i}$ and $S_{i}$-parameter are given by:(1)$$K_{i}=\frac{1}{\sqrt{Z_{i-1}^{\prime }Z_{i}^{\prime }}}\;\mathrm{and}\;i=1,\ldots ,n+1,$$
(2)$$S_{11,i}=S_{22,i}=\frac{K_{i}/Z_{0}-Z_{0}/K_{i}}{K_{i}/Z_{0}+Z_{0}/K_{i}}e^{-j\theta_{e}},$$
(3)$$S_{12,i}=S_{21,i}=\frac{2}{{j(K}_{i}/Z_{0}+Z_{0}/K_{i})}e^{-j\theta_{e}},$$
where *n* corresponds to the order of the filter, $Z_{i}^{\prime }=Z_{i}$ if *i* is odd or $Z_{i}^{\prime }=1/Z_{i}$ if *i* is even, and $Z_{0}=Z_{S}=Z_{L}=1$.

Table 3 gives the element values ($K_{i}$, $\left|S_{21,i}\right|$) of the six sections of the equivalent circuit (Figure 5b), which were obtained from the stepped impedance Chebyshev synthesis method and for the design specifications given in Table 1. All sections are selected to have the same phase shift value: ${∠S_{21,j}=-\theta }_{e}-90{}^{\circ}=-120{}^{\circ}$. The physical $t_{yi}$ and $L_{i}$ dimensions of each section are optimized by means of an EM simulator (HFSS) to match the $\left|S_{21,i}\right|$-magnitude and $∠S_{21,i}$-phase values included in Table 3. For this purpose, the same GGW dimensions included in Table 2 are used, and the widths $t_{xi}$ and $t_{zi}$ of the metallic pins are set at 2 mm. The GGW filter is built with the different previously designed sections and a final optimization is performed to compensate for the couplings. Table 4 includes the final dimensions of the fifth-order GGW Chebyshev filter shown in Figure 1 obtained from this design approach to achieve the design specifications given in Table 1. The size of the filter is: 47.48 mm (length) × 47.05 mm (width) × 9.525 mm (height).

Figure 6 depicts the EM and theoretical frequency responses of the fifth-order wideband GGW Chebyshev bandpass filter (Figure 1 and Figure 5b). The structure of the practical GGW filter (Figure 1) was simulated (HFSS) with the dimensions given in Table 2 and Table 4, while data included in Table 3 were used for the equivalent circuit model of Figure 5b. The conductors were considered perfect in the EM simulations. As can be seen in Figure 6, the EM simulations display wideband properties with a bandwidth *BW* = 3.7 GHz (*FBW* = 36.4%), which meets the design specifications initially defined in Table 1. This wideband bandpass response is due to the combination of the high-pass behavior of the GGW and the low-pass characteristics of the metallic pins and GGW sections of electrical length (*θ_e_*) that work as impedance inverters. In this way, the lower cut-off frequency *f_C*,*L_* = 8.3 GHz of this filter is consequence of the high-pass behavior of the GGW, while the higher cut-off frequency *f_C*,*H_* = 12 GHz is due to the low-pass characteristics of the metallic pins and GGW sections of electrical length (*θ_e_*). Return loss in the passband is less than *RL* = 20 dB, except at frequencies near to *f_C*,*L_*, where it is better than 10 dB. This deterioration is due to the high dispersion of the quasi-TE_10_ mode close to the GGW cut-off frequency (*f_C*,*L_*) and to the strong frequency dependence of the impedance inverters in this region [19,47]. The passband shows five reflection zeros, while the rejection band has a deep spurious-free range up to 15 GHz with a maximum rejection level *RLSB* = 77 dB. Outside the frequency range (8 GHz–15 GHz), the two periodic structures of the GGW no longer behave as PMC sidewalls (Figure 2) and leakage waves occur in the transverse directions between the gap *h* (Figure 1). As a result, spurious bands appear below 8 GHz and above 15 GHz. A comparison between theoretical and EM |*S*_21_| results (Figure 6) shows that the out-of-band frequency responses do not match. The out-of-band rejection of the theoretical results is worse than the practical GGW filter. This is because the six impedance inverters of the practical GGW filter (Figure 1) are realized by means of six metallic pins dependent of frequency, while the impedance inverters of the equivalent circuit (Figure 5b) are ideal and constant with the frequency. The six ideal inverters lead to a fifth-order Chebyshev response, while the metallic pins add six additional reactive elements in the frequency response of the practical GGW filter. As a results, the GGW structure shown in Figure 1 has a frequency response of an eleventh-order (5 + 6) filter, although this was determined from a fifth-order equivalent circuit [19,47].

## 3. Fabrication

The CNC (micro)machining technique is typically used to manufacture waveguide components at mm/microwave-wave frequencies. This technique requires an operator to carry out the machining of the device (choice of tool, rotation speed, etc.). The 3D printing technique is an emerging technology that is easier to use than the traditional machining method, since it does not require monitoring of any manufacturing process once the printing has started. Furthermore, 3D printing has recently been applied to GGW and RW technologies to produce low cost and weight plastic components [36,41,50]. These components, unlike those obtained by metal-based machining, have a rougher surface, and require metallization.

Three GGW Chebyshev bandpass filter prototypes have been manufactured to make a comparison between both manufacturing techniques. One was made in an aluminum material (Figure 7) using a CNC machining technique and the other two were obtained by means of a stereolithography-based 3D printer and a photopolymer resin (Figure 8). As can be seen in Figure 7 and Figure 8, all three prototypes are manufactured in two pieces (top and bottom parts) with the same horizontal cut. One of the two resin-based prototypes was metallized using an expensive high-vacuum evaporation technique, while the second one was coated by means of a cheap conductive paint spray system. The surface roughness of the three prototypes was measured with an optical microprofilometer (Taylor Hobson Talysurf CLI 500) in the same 2 mm^2^ area of the bottom part of the filter.

### 3.1. CNC Machining Technique

The CNC machining technique is a subtractive manufacturing method. This technique removes parts of a material bulk from rotary tools to achieve the desired device shape. The bulk of material can be plastic, wood, metal, etc. The two parts of the proposed GGW bandpass filter were CNC machined from a bulk of 6082 aluminum with a tolerance of ± 100 μm. The conductivity of 6082 aluminum given by the manufacturer is $24\;\times \;10^{6}\leq \sigma (S/m)\leq 32\;\times \;10^{6}$. Figure 7 shows the photograph of the two parts of the fifth-order GGW Chebyshev bandpass filter manufactured in aluminum using a CNC milling machine. Sidewalls and rectangular waveguide (RW) transitions have been added to the structure for the measurements. The sidewalls were located at 4 mm from the last row of pins of the periodic structure, while the rectangular waveguide transitions with WR-75 standard and length 5 mm were placed at 1.7 mm from the GGW bandpass filter input/output ports. An arithmetic average roughness of *R_a_* = 0.15 μm was measured.

### 3.2. 3D Printing Technique

The 3D printing technique is an additive method that consists of making the desired 3D device layer by layer by means of resins, plastic filaments, metallic powders, etc. This technology is more ethical than previous CNC machining, since it generates less material waste to produce devices. The two parts of the proposed GGW bandpass filter (Figure 8a) were realized from a stereolithography-based 3D printer (Form 3B, Formalabs) and a photopolymer resin (Clear Surgical Guide Resin). This printer can achieve a layer thickness (*y*-axis) of 50 μm and a resolution in the (*x*, *z*) plane of 25 μm from this resin. In the same way as for the previously CNC machined filter, sidewalls, and rectangular transitions (WR-75) have been added to the structure for the measurements. Two resin-based prototypes were made. The manufacturing process of the resin-based GGW bandpass filter finishes by coating their surfaces with a conductive layer. The surfaces of one resin-based prototype were metallized using a high-vacuum thermal evaporation deposition technique, while those of the second resin-based prototype were coated by means of a conductive paint spray system.

#### 3.2.1. High-Vacuum Evaporation Technique

A high-vacuum thermal evaporation deposition technique was applied to the metallization of the two parts of one of the two resin-based prototypes. It is a method of thin-film deposition. An in-house high-vacuum evaporation system (Figure 9) was used. It consists of placing one of the two parts of the resin-based prototype (Figure 8a) inside the vacuum chamber, creating the vacuum by means of primary and turbomolecular pumps, and coating their surfaces through the thermal evaporation of a metal (Au, Ag, Cu, etc.) wire introduced inside an electric tungsten (W) filament. This process was also carried out for the second part of the resin-based prototype. A silver layer ($\sigma =63\;\times \;10^{6}\;S/m)$ of thickness *t* = 5 μm was deposited on the surfaces of the two parts of the resin-based GGW bandpass filter prototype (Figure 8b) using this in-house high-vacuum evaporation system (Figure 9). An arithmetic average roughness of *R_a_* = 1 μm was obtained.

#### 3.2.2. Conductive Paint Spray System

This method consists of an impregnation technique that allows the deposition of thick films by spray coating. This technique is cheaper and faster than the previous one since it does not require expensive equipment or a long vacuum process. A conductive layer ($\sigma =10^{4}\;S/m$) of thickness *t* = 50 μm was deposited on the surfaces of the two parts of the resin-based GGW bandpass filter prototype (Figure 8c) using a conductive paint spray system (Figure 10). The spray system contained the MG Chemicals 843AR conductive paint, which is composed of acrylic lacquer, pigmented with silver-coated copper flakes. The arithmetic average roughness of this metallized resin-based prototype is *R_a_* = 4.8 μm.

## 4. Experimental Results

The measured and EM simulation frequency responses of the three manufactured GGW bandpass filter prototypes shown in Figure 7 and Figure 8b,c are plotted in Figure 11, Figure 12 and Figure 13, respectively. The prototypes were measured from a vector network analyzer (R&S ZVA67). A WR-75 TRL (through, reflect, line) calibration KIT and a TRL calibration procedure were used for de-embedding the effects of the cables and SMA-rectangular waveguide transitions from the measurements of the prototypes between 8 GHz and 15 GHs. In this way, the *S*-parameters measurements of the prototypes were obtained between their input and output ports. This frequency range (8 GHz–15 GHz) allows single-mode operation (TE_10_) in the WR-75 rectangular waveguide and GGW (Figure 2). Below 8 GHz and above 15 GHz, accurate measurements of the prototypes cannot be carried out, since calibration errors and power fluctuations occur due to the cut-off frequency of the WR-75 rectangular waveguide TE_10_ mode and the superposition of the first- and higher-order modes in their structures (Figure 2), respectively. The conductivity (σ) of the conductor and the roughness ($R_{a}$) of the surfaces were considered in the EM simulations (HFSS) of these three prototypes.

The measured *S*-parameters (Figure 11) of the fifth-order GGW Chebyshev bandpass filter manufactured in aluminum using a CNC milling machine show a good agreement with the simulated data. Regarding the EM simulations, the measurements show a slight shift of 50 MHz towards higher frequencies due to the manufacturing tolerance, which is *±*100 μm. Between 10 GHz and 12 GHz, corresponding to the lower frequency of the recommended band of the WR-75 standard and the higher cut-off frequency *f*_*c*,*H*_ of the filter, the return and insertion losses are, respectively, *RL* = 20 dB and *IL* = 0.3 dB. These losses deteriorate between 8.3 GHz and 10 GHz, due to the high dispersion of the quasi-TE_10_ mode (Figure 2) and to the frequency dependence of the impedance inverters (metallic pins) in this frequency range [19,47]. The out-of-band region has a spurious free range up to 15 GHz with a maximum rejection level *RLS*B = 70 dB.

The resin-based prototypes (Figure 12 and Figure 13) exhibit a bandwidth *BW* = 3.7 GHz and a stopband with a spurious free range up to 15 GHz like the previous aluminum-based GGW bandpass filter. The measured results of both resin-based prototypes agree well with the simulated data, although some differences in return loss and number of reflection zeros can be seen. The measured return and insertion losses in the frequency range 10 GHz–12 GHz for the resin-based filter (Figure 12) metallized with a silver layer are *RL* = 20 dB and *IL* = 1 dB, respectively. Its maximum rejection level in the stopband is *RLSB* = 65 dB. Except for *RL* = 15 dB, the resin-based filter (Figure 13) metallized with a conductive paint spray has similar results: *IL* = 1.2 dB and *RLSB* = 71 dB.

A comparison between machined (Figure 11) and 3D printed (Figure 12 and Figure 13) prototypes shows both resin-based filters present worse insertion loss than the aluminum-based one. This is because the surface roughness of the 3D printed prototypes has a negative effect on the conductor loss of the silver and conductive paint layers. The roughness of the metallized surfaces is larger than the one obtained with the CNC technique, and consequently the attenuation increases [51]. In addition, the conductive paint has a poor conductivity ($\sigma =10^{4}\;S/m)$, which is approximately 6300 and 2800 times less than the value of the silver and 6082 aluminum, respectively. The other measured characteristics of both resin-based filters are similar to aluminum-based prototype and initial design specifications (Table 1). The advantage of the 80 g resin-based prototype is that it is 2.6 times lighter than its aluminum-based one (208 g).

Table 5 compares various GGW bandpass filters with the wideband GGW Chebyshev filters manufactured in this work. The order *n* or number of cells of the GGW filters [30,31,32,33,34,35,36,37,38,39,40,41,42,43,44,45] reported in Table 5 varies between two and seven, and almost half of them are of order five, as are the proposed GGW filters. The structures were mainly manufactured by means of a CNC milling machine, except for two [36,41] that also used a 3D printing technique. The different designs cover the bands X to W (8 GHz–110 GHz). Half of them are distributed in the low X and Ku bands (8 GHz–18 GHz) as the proposed design. As can be seen in Table 5, unlike the proposed design method and [45], which can achieve wideband responses (*FBW* > 10%), the other filters developed are narrowband (*FBW* < 4%), since they were obtained from resonant cavities [30,31,32,33,34,35,36], coupled resonators [37,38,39,40,41,42], evanescent wave coupled resonators [43], and periodic corrugations acting as resonators [44], which can only work in a narrow bandwidth. The wideband response of the proposed GGW bandpass filter design was obtained by combining the high-pass behavior of the GGW with the low-pass characteristic of the metallic pins placed along the groove area of the GGW, acting as impedance inverters. Contrary to [45], a rigorous synthesis method based on stepped impedances was applied to the metallic pins to obtain the desired higher cut-off frequency ($f_{c,H}$) of the passband. The three manufactured wideband GGW bandpass prototypes present high performance in the passband and rejection band with respect to the other GGW filters. Except the resin-based GGW prototype metallized with a conductive paint spray (Figure 13) that has *RL* = 15 dB and *IL* = 1.2 dB, the other two wideband GGW filters (Figure 11 and Figure 12), based on aluminum and resin coated with a silver layer, exhibit better *RL* and *IL* than most of the filters reported in Table 5. In addition, the maximum level reached in the stopband (*RLSB*) is similar to the highest values obtained with the narrowband filters. The three fabricated prototypes did not require any post-manufacturing tuning, since the sensitivity to the mechanical tolerance is lower than bandpass filters based on high-*Q* resonators. It should also be noted that the metallic pins along the groove area of the GGW, which are properly centered, distributed, and sized according to a stepped impedance technique, allow the implementing of compact wideband Chebyshev GGW filters. The designed fifth-order wideband GGW filter has a first-order normalized area of $0.53\lambda_{0}^{2}$. In Table 5, this area is, respectively, 1.4 and 9.9 times less than the smallest [45] and largest [33] GGW bandpass filter.

## 5. Conclusions

A fifth-order wideband Chebyshev bandpass filter in groove gap waveguide (GGW) technology is proposed for the first time in this paper. This filter is conceived from metallic pins, acting as impedance inverters. The metallic pins are adequately centered, distributed, and sized along the groove area of the GGW, according to a Chebyshev stepped impedance technique. The bandpass response of this filter is obtained by combining the high-pass behavior of the GGW with the low-pass characteristics of the metallic pins. This design technique is simple and allows suitable implementations of high-performance wideband filters using a computer numerical control (CNC) machining or an emerging three-dimensional (3D) printing technique, thus providing low-cost and low-weight prototypes. Three prototypes of the proposed filter have been fabricated: one using a CNC milling machine and the other two by means of a stereolithography-based 3D printer and a photopolymer resin. One of the two resin-based prototypes has been metallized using an expensive high-vacuum silver evaporation technique, while the other has been coated by means of a cheap conductive paint spray system. The measured frequency responses of these prototypes have shown good agreement with EM simulations. The proposed filter is less sensitive to mechanical tolerance since it is not based on high-*Q* resonators. The three prototypes have exhibited high-performance responses and a reduced size compared to other GGW bandpass filters. These wideband GGW filter prototypes are expected to be useful in future emerging communications systems.

## Figures and Tables

**Figure 1 sensors-23-06234-f001:**
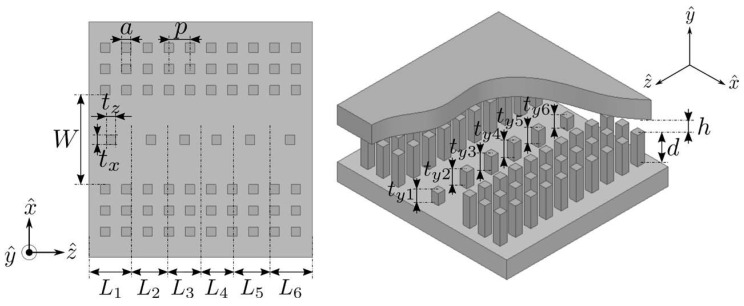
Structure of the wideband Chebyshev bandpass filter based on GGW technology (with order *n* = 5).

**Figure 2 sensors-23-06234-f002:**
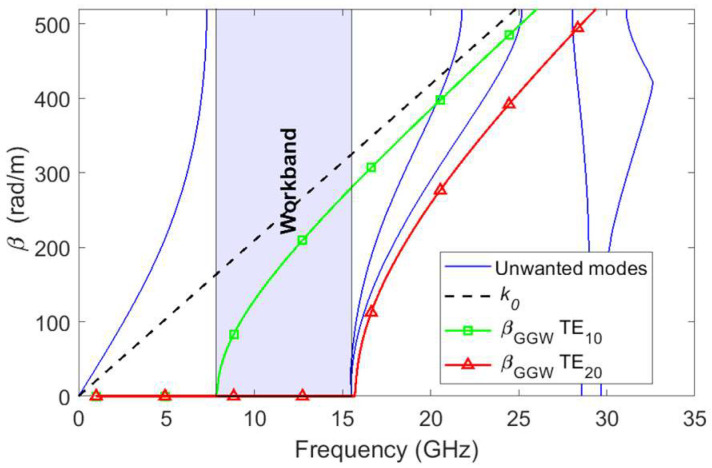
Dispersion diagram for the GGW dimensions defined in Table 2.

**Figure 3 sensors-23-06234-f003:**
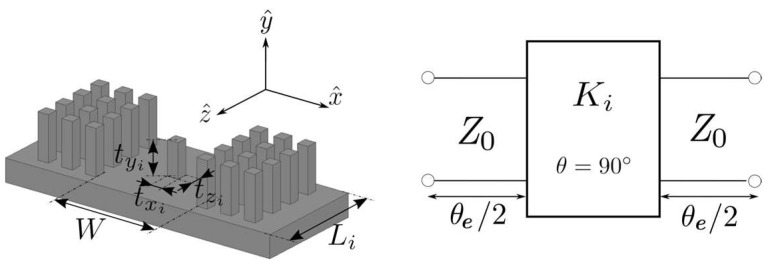
Structure (without the upper plate) and equivalent circuit of the *i*th-section of the GGW filter shown in Figure 1.

**Figure 4 sensors-23-06234-f004:**
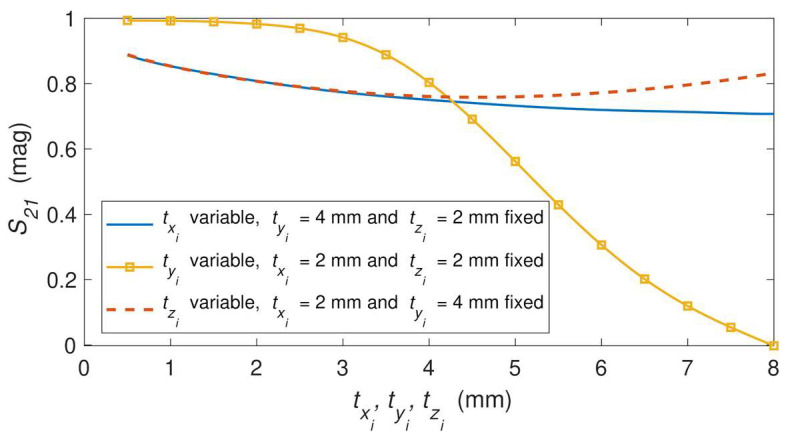
|*S*_21_|of the simulated *i*th-section (Figure 3) as a function of the dimensions of the metallic pins: *t_xi_* (*t_zi_* = 2 mm and *t_yi_* = 4 mm fixed), *t_yi_* (*t_xi_* = *t_zi_* = 2 mm fixed) and *t_zi_* (*t_xi_* = 2 mm and *t_yi_* = 4 mm fixed).

**Figure 5 sensors-23-06234-f005:**
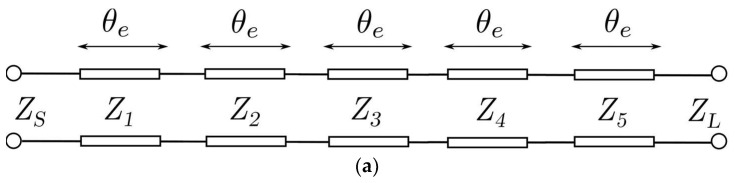
Equivalent circuits of a fifth-order stepped impedance filter. (**a**) Implementation using stepped impedance transmission lines. (**b**) Implementation using impedance inverters and transmission lines with the same impedance and electrical length.

**Figure 6 sensors-23-06234-f006:**
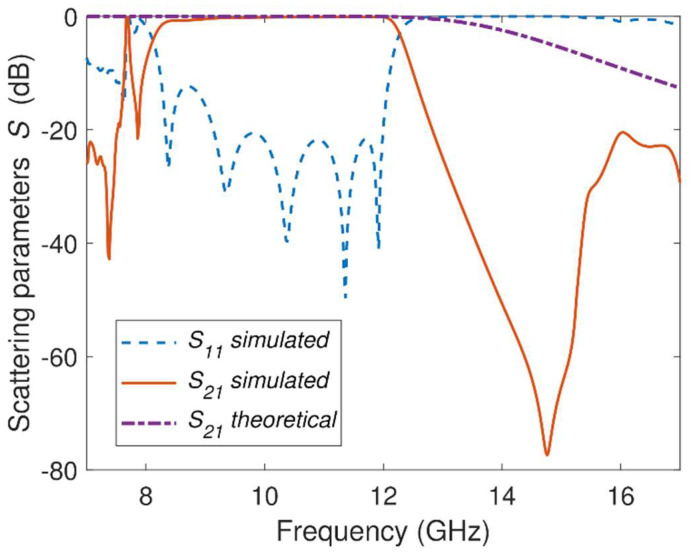
Theoretical and simulated *S*-parameters of the fifth-order wideband GGW Chebyshev bandpass filter.

**Figure 7 sensors-23-06234-f007:**
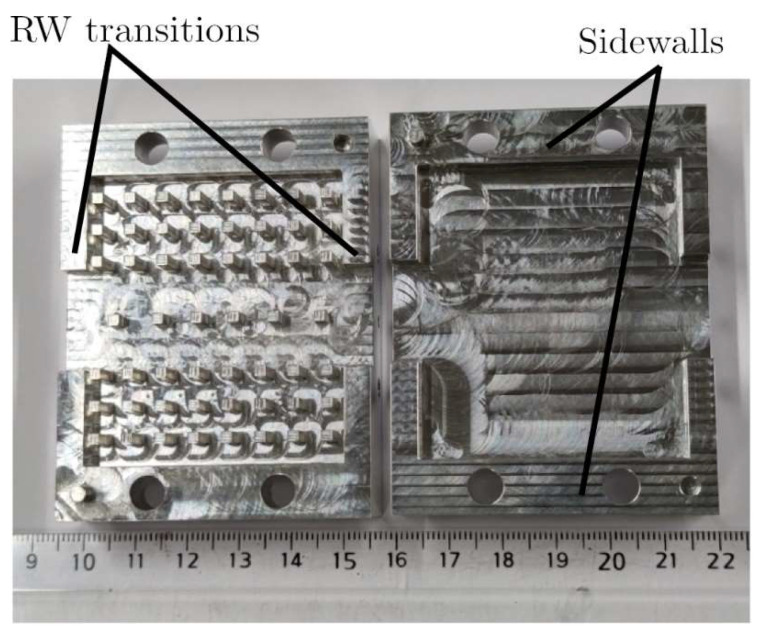
Unassembled fifth-order wideband GGW Chebyshev filter manufactured in aluminum using a CNC machining.

**Figure 8 sensors-23-06234-f008:**
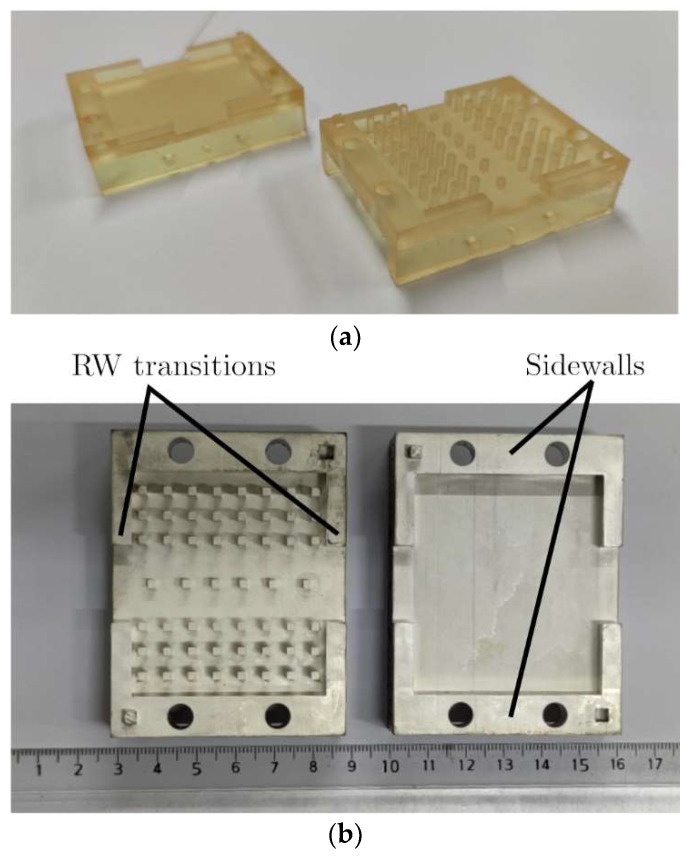
Unassembled fifth-order wideband GGW Chebyshev filters manufactured with a stereolithography-based 3D printer and a resin. (**a**) Prototype without metallization. (**b**) Silver metallized prototype using a high-vacuum evaporation technique. (**c**) Conductive paint metallized prototype using a spray system.

**Figure 9 sensors-23-06234-f009:**
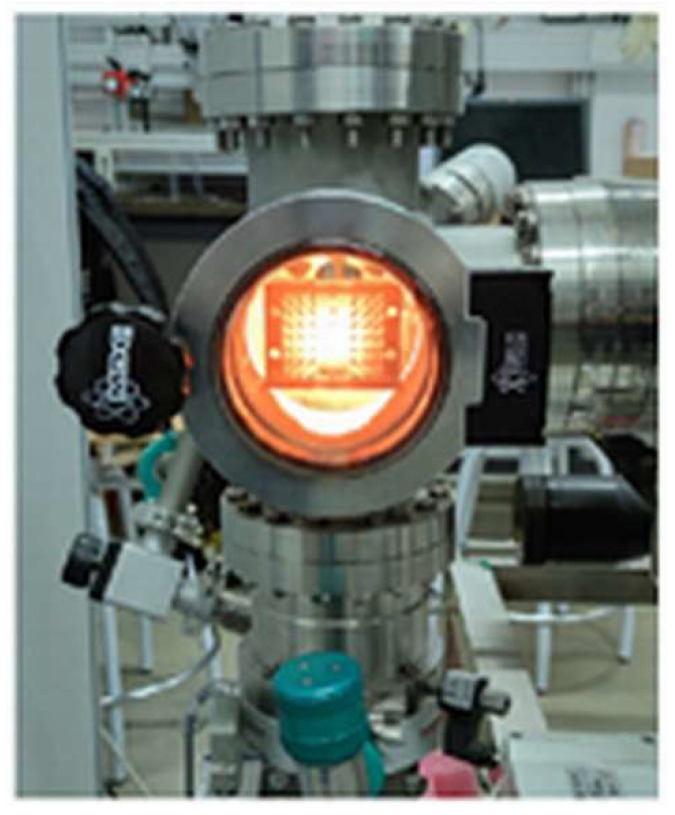
Photograph of the in-house high-vacuum evaporation system loaded with the bottom part of the polymerized resin-based GGW filter prototype during the metallization process.

**Figure 10 sensors-23-06234-f010:**
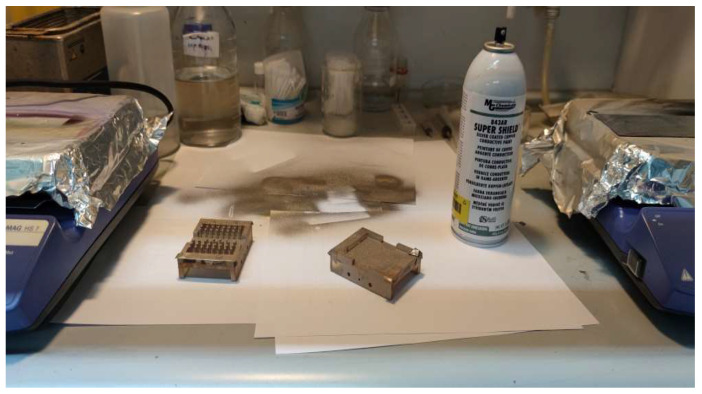
Photograph of the two parts of the conductive paint metallized resin-based GGW filter prototype using a spray system.

**Figure 11 sensors-23-06234-f011:**
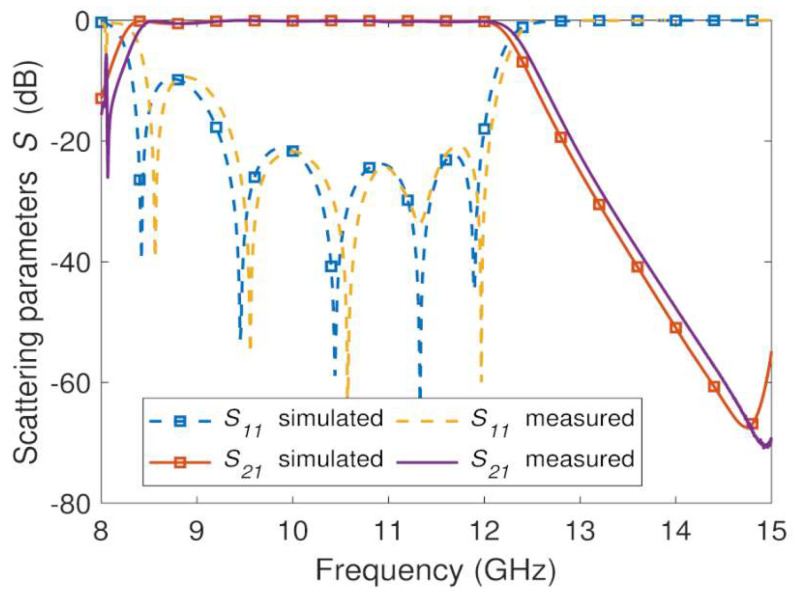
EM simulated and measured *S*-parameters for the aluminum fifth-order wideband GGW Chebyshev filter (Figure 7).

**Figure 12 sensors-23-06234-f012:**
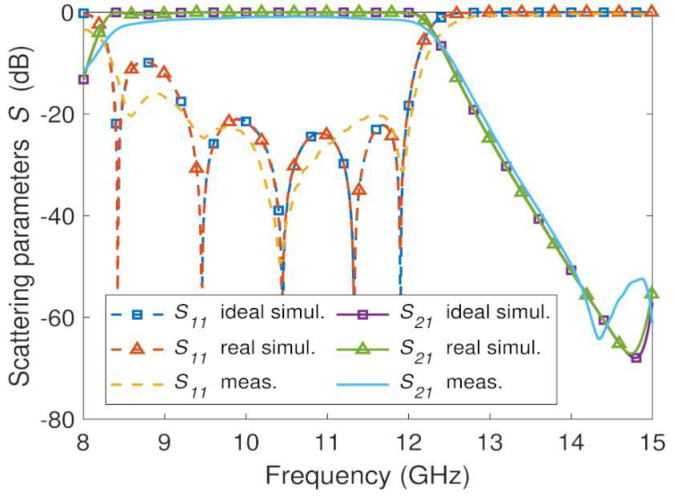
EM simulated and measured *S*-parameters for the silver metallized resin-based fifth-order wideband GGW Chebyshev filter (Figure 8b).

**Figure 13 sensors-23-06234-f013:**
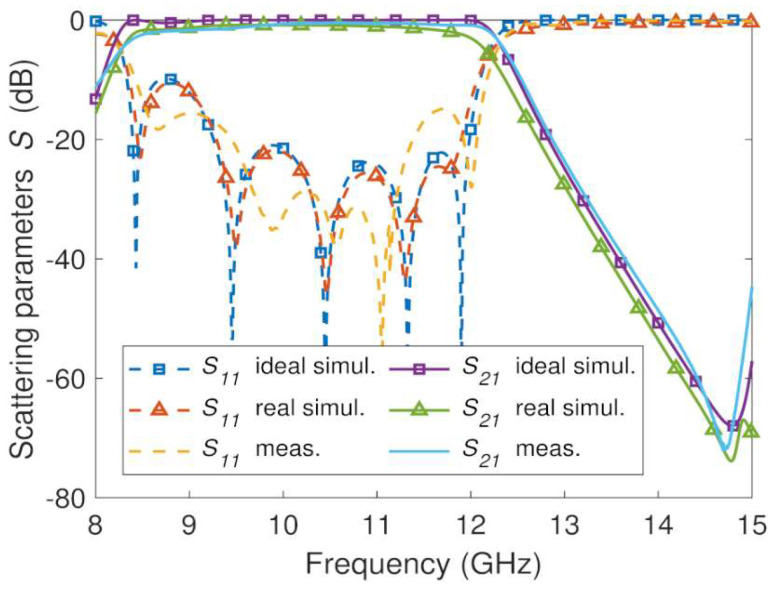
EM simulated and measured *S*-parameters for the conductive paint metallized resin-based fifth-order wideband GGW Chebyshev filter (Figure 8c).

**Table 1 sensors-23-06234-t001:** Design specifications for the GGW bandpass filter shown in Figure 1.

Parameter	Value
Type	Chebyshev
Order, *n*	5
Return loss, *RL* (dB)	20
GGW TE_10_ mode cut-off frequency, *f_c*,*L_* (GHz)	8.3
Higher cut-off frequency at |*S*_11_| = −20 dB, *f_c*,*H_* (GHz)	12
Cut-off angle, *θ_e_* (°)	30

**Table 2 sensors-23-06234-t002:** Optimized GGW dimensions.

Parameter	Value (mm)	Parameter	Value (mm)
*d*	8	*p*	6
*a*	2	*W*	19.05
*h*	1.525	*b*	9.525

**Table 3 sensors-23-06234-t003:** Impedance inverter *K*, |*S*_21_| and ∠*S*_21_ values extracted from the stepped impedance Chebyshev synthesis method for the design specifications included in Table 1.

Parameter	Value	Parameter	Value
*K*_1_ = *K*_6_	0.704	|*S*_21,1_| = |*S*_21,6_|	0.941
*K*_2_ = *K*_5_	0.457	|*S*_21,2_| = |*S*_21,5_|	0.756
*K*_3_ = *K*_4_	0.364	|*S*_21,3_| = |*S*_21,4_|	0.643
		∠*S*_21,*i*_ *i* = 1, …, 6	−120°

**Table 4 sensors-23-06234-t004:** Dimensions of the fifth-order wideband GGW Chebyshev filter (Figure 1) for the design specifications included in Table 1.

Parameter	Value (mm)	Parameter	Value (mm)
*t_y_*_1_ = *t_y_*_6_	3.09	*L*_1_ = *L*_6_	8.985
*t_y_*_2_ = *t_y_*_5_	4.19	*L*_2_ = *L*_5_	7.835
*t_y_*_3_ = *t_y_*_4_	4.56	*L*_3_ = *L*_4_	6.96
*t_xi_* = *t_zi_ i* = 1, …, 6	2		

**Table 5 sensors-23-06234-t005:** Comparison between the manufactured fifth-order wideband GGW Chebyshev filters and other GGW bandpass filters reported in the literature.

Ref.	Manufacturing/ Material + Plating	Order *n*/ Cells	*f_o_* (GHz)	*FBW* (%)	*RL* (dB)	*IL* (dB)	*RLSB* (dB)	*Q_u_*	2-D Size (*λ*_0_ × *λ*_0_)
[30] Figure 11b	CNC machining/ aluminum	5	14	1	13	1	80	3480	5.1 × 1.13
[31] Figure 9	CNC machining/ aluminum	4	40	2.5	19	1.1	32	1005	4.1 × 1
[32] Figure 5	CNC machining/ brass + silver	5	59.7	1.34	9	1.7	55	1046	6.92 × 2.43
[33] Figure 9	CNC machining/ aluminum	5	11.58	0.62	10	0.9	60	-	6 × 4.4
[34] Figure 11	CNC machining/ aluminum	5	11.59	0.86	19	0.7	55	5402	8 × 1.8
[35] Figure 5	CNC machining/ aluminum	3	16	2	21.2	0.53	40	1560	6.2 × 1.7
[36] Figure 7	3D printing/ resin + silver	5	94	3.2	15	0.46	50	2210	10.2 × 1.8
[37] Figure 15	CNC machining/ aluminum	7	37.3	1.8	17	1.5	70	1691	9.62 × 1.48
[38] Figure 7	CNC machining/ aluminum	5	61	2.5	13	1.5	50	928	4.54 × 1.16
[39] Figure 18	CNC machining/ brass + silver	3	35	1	9	1	35	1091	1.48 × 2.08
[40] Figure 11	CNC machining/ unknown	5	12.08	0.5	15	2	70	44	5.12 × 1.37
[41] Figure 7	3D printing/ polymer + copper	4	35.65	1.4	11.7	0.5	48	1275	2.5 × 2.5
[41] Figure 9	CNC machining/ brass	4	36	1.4	11	1.2	52	614	2.5 × 2.5
[42] Figure 7	CNC machining/ aluminum	2	16	3.63	15	1.7	55	-	-
[43] Figure 10	CNC machining/ aluminum	4	12.1	2.4	20	0.3	83	3490	2.23 × 1.65
[44] Figure 16	CNC machining/ aluminum	6	11.87	3.88	18	0.5	75	2152	2.31 × 3.38
[45] Figure 7	CNC machining/ aluminum	6	27	22.2	17.3	0.8	30	-	2.35 × 2
Figure 11	CNC machining/ aluminum	5	10.2	36.2	20	0.3	70	299	1.61 × 1.62
Figure 12	3D printing/ resin + silver	5	10.1	36.3	20	1	64	89	1.61 × 1.62
Figure 13	3D printing/ resin + conductive paint	5	10.1	36.8	15	1.2	71	73	1.61 × 1.62

where *f*_0_, *FBW*, *RL*, *IL*, *RLSB, Q_u_* and *λ*_0_ are, respectively, the central frequency, fractional bandwidth, return loss, insertion loss, maximum rejection level in the stop-band regions, unloaded quality factor, and free-space wavelength at *f*_0_.

## Data Availability

Not applicable.

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
