# Peer review of "Compact Wideband Groove Gap Waveguide Bandpass Filters Manufactured with 3D Printing and CNC Milling Techniques"

_sensors, 2023, doi:10.3390/s23136234_

Round 1

Reviewer 1 Report

The authors present a fifth-order Chebychev wideband bandpass filter in groove gap waveguide technology at approx. 10 GHz. They present the design guidelines, the simulations and three prototypes manufactured by different technologies, which are experimentally tested and compared between them and with other works. The work is very complete and interesting.

In Fig. 6, the circuit response corresponding to an ideal Chebychev filter could be added, to have a comparison.

At some point, it would be interesting to have a response wider than that presented, for the simulated filter and for the experimental results, to clearly see the out of band response. At the lower end, cutoff frequency for the WR75 rectangular waveguide transitions is very close to the bandpass, and authors could comment on the calibration issues. At the upper end of the band, it seems that the structure is symmetric with respect to the yz-plane, so there should be margin to plot the response since TE20-like modes would not be excited.

Comparison in Table 5 adds a lot of insight into the problem. It is very interesting to see the insertion losses of the three prototypes. Moreover, the extracted equivalent unloaded Q for comparing with other filters gives a lot of information. Perhaps, given the frequency band addressed in the paper, traditional ridge waveguide would also play an important role for wideband filters. Moreover, authors could comment more on the benefits of using groove gap waveguide technology depending on the frequency band, taking into account the size, complexity and fabrication costs associated to the pins with respect to classic waveguide technologies at lower bands.

Author Response

Dear Editor Dr. Winston Wang and Reviewer 1,

Please find in the following our answers to all the suggestions of the reviewer 1. The changes in the revised version of the manuscript are highlighted in blue and green according to the recommendations of the reviewers 1 and 2, respectively.

Best regards,

Dr. Juan Hinojosa.

Reviewer 2 Report

A compact wideband bandpass filter in groove gap waveguide (GGW) technology is proposed in this work. Here are my comments

1-      Why did the authors choose the filter with 5th order

2-      Figures  with high resolution should be added

3-      Why does the cut-off frequency at the lower band has a low value at 8 GHz which reduces the selectivity at the lower frequency band

4-      How can you control the bandwidth and transmission zeros in this design

5-      More parametric studies should be added

6-      New references from 2022-2023 should be added to the table of comparison.

 Moderate editing of English language required

Author Response

Dear Editor Dr. Winston Wang and Reviewer 2,

Please find in the following our answers to all the suggestions of the reviewer 2. The changes in the revised version of the manuscript are highlighted in blue and green according to the recommendations of the reviewers 1 and 2, respectively.

Best regards,

Dr. Juan Hinojosa.

Round 2

Reviewer 2 Report

The author's replied to all comments. Best Regards,